# Factors motivating female sex workers to initiate pre- exposure prophylaxis for HIV prevention in Zimbabwe

**Definate Nhamo**[1]*, **Sinegugu E. Duma**[1], **Elizabeth B. Ojewole**[2], **Dixon Chibanda**[3], **Frances M. Cowan**[4,5]

**1** Department of Nursing and Public Health, University of KwaZulu-Natal, Durban, South Africa, **2** Discipline of Pharmaceutical Sciences, College of Health Sciences, University of KwaZulu-Natal, Durban, South Africa, **3** Department of psychiatry, University of Zimbabwe, Harare, Zimbabwe, **4** Centre for Sexual Health and HIV Research (CeSHHAR) Zimbabwe, Harare, Zimbabwe, **5** Department of International Public Health, Liverpool School of Tropical Medicine (LSTM), Liverpool, United Kingdom

* defnh78@gmail.com

## Abstract

**Editor:** Hamid Sharifi, HIV/STI Surveillance Research Center and WHO Collaborating Center for HIV Surveillance, Institute for Future Studies in Health, Kerman University of Medical Sciences, ISLAMIC REPUBLIC OF IRAN

### Background

Female sex workers (FSWs) have a 26 times greater chance of HIV infection compared to the women in the general population. The World Health Organization recommends pre-exposure prophylaxis (PrEP) for population groups with an HIV incidence of 3% or higher and FSWs in southern Africa fit this criteria. This study sought to understand factors that motivate FSWs to initiate PrEP, in Harare, Zimbabwe.

### Methods

We purposively selected and recruited 20 FSWs to participate in the study in-order to gain an in-depth understanding of factors that motivate FSWs to initiate PrEP in Harare, Zimbabwe. We identified FSW who had been initiated on PrEP at a specialized clinic providing comprehensive sexual reproductive health (SRH) services for sex workers including HIV prevention options. We used a descriptive phenomenological approach to collect and analyze the data. Data was analyzed using Colaizzi's seven steps to analyze data.

### Findings

Two broad themes were identified as intrinsic and extrinsic motivators. The two broad themes each have several sub-themes. The sub-themes under intrinsic motivation were (i) Self- protection from HIV infection and (ii) condoms bursting. Six sub-themes were identified as external motivators for initiating PrEP, these included (i) occupational risk associated with sex work, (ii) increased chance of offering unprotected sex as a motivator to initiate PrEP, (iii) positive encouragement from others (iv) need to take care of the children and (v) prior participation in HIV prevention research studies and (vi) Gender Based Violence.

**Data Availability Statement:** All relevant data are within the manuscript and its Supporting Information files.

**Funding:** This work was supported through the DELTAS Africa Initiative [DEL-15-01]. The DELTAS Africa Initiative is an independent funding scheme of the African Academy of Sciences (AAS)'s Alliance for Accelerating Excellence in Science in Africa (AESA) and supported by the New Partnership for Africa's Development Planning and Coordinating Agency (NEPAD Agency) with funding from the Wellcome Trust [DEL-15-01] and the UK government. The views expressed in this publication are those of the author(s) and not necessarily those of AAS, NEPAD Agency, Wellcome Trust or the UK government. IREX funding through the UASP alumni small grants program is acknowledged for providing a platform for the development of this publication.

**Competing interests:** The authors have declared that no competing interests exist.

## Conclusions

Understanding the factors that motivate FSWs to initiate PrEP is critical in developing contextually appropriate strategies to promote PrEP initiation and adherence strategies within specific and eligible populations for receiving PrEP according to the WHO guidelines (2015).

## Introduction

Female sex workers (FSWs) are 26 times more likely to be infected with HIV compared to women in the general population [1,2]. Nine percent of the global HIV infections occur within this population [3]. Pre-Exposure Prophylaxis (PrEP) is the use of antiretroviral drugs to prevent HIV infection [4]. The World Health Organization (WHO) recommends PrEP to prevent HIV infection (2015) and that PrEP should be offered to all population groups at substantial risk of incident infection (defined as 3% or higher). Female sex workers (FSW) in Zimbabwe are very vulnerable to HIV infection with an estimated HIV incidence of 10% against a national HIV incidence of 0.47% [5–8]. Other groups identified as vulnerable include sero-discordant couples (SDC), pregnant and breastfeeding women (PBFW), men who have sex with men (MSM) and gay and bisexual men (GBM) [1].

Several studies have been conducted to explore the motivations for PrEP initiation across different population groups including a Kenyan study among pregnant and breastfeeding women [9]; the Thika study in Kenya on sero-discordant couples [10]; Asian and USA-based studies among gay and bisexual men [11,12], and a Zimbabwean study conducted in the general population [13]. These studies highlighted that different populations are motivated by different factors. The sero-discordant couples study underscored PrEP as an additional strategy to reduce the risk of HIV infection to the HIV negative partner whilst meeting their fertility desires [10]. In addition, sero-discordant couples highlighted the need for provider confidence and accessing client-friendly services as motivating factors for initiating on PrEP. Pregnant and breastfeeding women identified previous STI diagnosis, forced sex, prior experience of intimate partner violence, prior use of post-exposure prophylaxis and a partner living with HIV as motivation for them to take PrEP [9]. In general population studies, motivation for PrEP ranged from partner's HIV status, confidence in sexual relationship, less stress associated with condom use and whether one had family or partner support [13]. Some barriers identified in these studies were logistical challenges associated with getting to the health facility and associated side effects PrEP [13]. Men who have sex with men highlighted that taking PrEP incited conflict within their relationships because of the perceptions that PrEP users engaged in risky behaviors. They also reported judgmental health providers and stigma associated with PrEP disclosure [14]. MSM further reported that their decisions to initiate PrEP took into account related side effects, burden of daily dosing, frequency of sexual encounters and the sexual freedom that PrEP came with [15].

It is clear that motivational factors differ invariably from one population to the other. Few studies have been conducted on what motivates FSWs to initiate PrEP, despite being a highly vulnerable and eligible group for accessing PrEP. Understanding the motivational factors for PrEP among FSWs may help in developing appropriate strategies to promote PrEP and effective PrEP adherence interventions for FSW. Lack of understanding of the factors that motivate FSWs to initiate PrEP may have huge public health and financial implications as new HIV infections will continue to filter into the general population [16,17]. This study explored the factors that motivate FSWs to initiate PrEP in Harare, Zimbabwe's capital city.

## Methods

### Setting

The study was conducted at a specialized clinic for FSWs situated in Mbare–one of the capital's oldest high-density suburbs. The clinic offers free comprehensive sexual reproductive health (SRH) services, including provision of oral PrEP to FSW from all over Harare [18].

### Design

The study employed a Descriptive Phenomenological Approach (DPA) to understand FSW motivations for initiating on PrEP. The DPA is a qualitative research approach that examines how people explore and decipher their significant life experiences [19]. This approach allows the researcher into the participants life of motivational factors for initiating on PrEP as explained by Giorgi, 2009 [20].

### Population and sampling technique

The study population consisted of all FSWs who were initiated on oral PrEP within a larger sexual reproductive health and HIV prevention programme delivering services through the specialized Mbare clinic. The sex workers routinely frequent the clinic for various sexual reproductive health (SRH), including HIV prevention services. Purposive sampling was used to select and recruit 20 women who self-identified as FSWs, had been initiated on PrEP and had visited the programme at the time. This predetermined sample size of 20 FSWs is in line with principles of a descriptive phenomenological approach (DPA). Twenty was deemed adequate to elicit an in-depth understanding of the FSW's perspectives on their motivations for initiating PrEP in Harare and not in generalizing the findings [21,22].

### Recruitment process

FSWs presenting at the clinic for their first PrEP follow up visit were approached and informed about the study and invited to participate. This was normally within four weeks of PrEP initiation. It was important to elicit their perspectives on the motivations for accessing PrEP at this stage before they experienced barriers to PrEP adherence. Those who were interested in joining the study were given an information sheet to read. This was followed by a verbal explanation of the study and some questions to assess comprehension. Thereafter, FSWs were given the informed consent form to read and subsequently sign, thus indicating their agreement to voluntary participation in the study. They were given a day to make an informed decision on whether or not they were interested in participating in the study. In line with the Medical Research Council of Zimbabwe (MRCZ), those who decided to be part of the study, returned to the clinic the following day for an interview and were given US$5 to reimburse them for transport and time spent at participating in the study [23].

### Data collection

Data was collected from March—August 2020 using the semi-structured interview guide. The English interview guide was translated into both Shona and Ndebele, Zimbabwe's two main local languages, by professional bi-lingual translators who were conversant with all local languages. The translated guides were checked by an independent researcher for any loss in meaning. Individual face-to-face interviews were conducted with each participant in a private and quiet space within the clinic. Interviews were either conducted in English, Ndebele or Shona, depending on the participant's language preference. Probing questions were asked to elicit a deeper understanding and get more nuanced descriptions of participant's motivations

for initiating on PrEP. Each interview was audio-recorded and translated within the first 24 hours. The first author conducted all the 20 interviews from March–August 2020. The first author received numerous trainings on qualitative research including how to conduct descriptive phenomenology and analysis. The first author was guided and supported by the second author who was the supervisor and Professor of qualitative research, with more than two decades of experience in conducting qualitative research.

**Interview guide.**    The interview guide was developed by the first author in consultation with the second author, who has more than two decades of experience in developing qualitative guides. The interview guide was developed to be responsive to the research objective which was aimed at exploring factors motivating female sex workers to initiate PrEP.

## Pilot study

Pilot interviews were conducted with the first three participants we recruited to the study. We used the same recruitment strategy and semi-structured interview guide designed for the main study. The objective of the pilot was to check whether the semi-structured interview guide questions were clear and could easily be understood by the participants. No changes were made to the guide as a result of the pilot. Data from the pilot study were included and analysed as part of the main study [24,25].

## Data analysis

Data analysis followed the seven descriptive phenomenological data analysis steps described by Colaizzi [26]. Firstly, the researcher familiarized themselves with the data through reading each of the 20 translated transcripts. Secondly, statements which made direct reference to the factors that motivate FSWs to initiate PrEP were identified and labeled as such. Thirdly, these were carefully considered for relevance and meaning. Fourthly, the researcher analysed and clustered all the statements to develop themes as findings of factors that motivate FSWs to initiate PrEP initiation. The fifth stage involved in-depth descriptions of the emerging themes. As a sixth step, the exhaustive descriptions were truncated to capture the essence of the phenomenon through short, dense statements deemed essential for the structure of motivational factors for PrEP initiation. Lastly, the researcher presented the short, dense statements from the process, to the FSWs for them to confirm whether or not the themes developed represented their views on the factors that motivate them to initiate PrEP through member checking [27]. In addition, each interview was transcribed within 24 hours of conducting the interview. Nvivo was used to manage the data, extract and classify FSW phrases. This made it easier to identify emerging themes across the FSW motivations as each additional FSW was recruited.

## Scientific rigor

A few strategies were applied to ensure credibility. Firstly, the raw data and the emerging themes were shared with a senior researcher, who is an expert in qualitative research (second author) to confirm if the themes developed were informed by the FSWs voices. Secondly, member checking was done to confirm that the developed themes were the true representation of the FSWs views on factors that motivate them to initiate PrEP. To ensure transferability, a thick description of themes was provided as excerpts of participant voices. An audit trail of all systematic processes used in conducting the study has been kept. Lastly, the researcher put aside any assumptions she had to fully understand what motivates FSWs to initiate PrEP objectively. All this was done through a process of bracketing, a qualitative approach used to minimize the potentially damaging effects of preconceptions that may taint the research process [28,29]. This was achieved through researcher reading and re-reading the transcripts to

fully engage with the data as well as through the use of direct quotations from the FSWs themselves to explain their motivations for initiating on PrEP. Member checking involves the taking of findings back to the study participants to ensure data has been interpreted correctly taking the context into account [29,30].

## Ethical considerations

Ethical clearance was obtained from the University of KwaZulu Natal's Biomedical Research Committee (BREC REF: **BREC/00000952/2020**), South Africa and the Medical Research Council of Zimbabwe (MRCZ /A/2534) prior to data collection. The FSWs signed the consent form confirming their agreement to voluntary participation in the study.

## Findings

**Demographic characteristics.** As shown in Table 1, twenty FSWs aged between 18 and 55 years participated in the study. Four had primary school level of education, 11 had secondary school whilst five had a college certificate. Fifteen reported being either separated or divorced, four were never married whilst one was cohabiting. The majority had an average of three children of their own.

The findings revealed two major themes namely intrinsic and extrinsic motivations. The two major themes each had several sub-themes. Intrinsic motivational factors had two sub-themes (self-protection from HIV infection and condoms bursting). Extrinsic motivational factors had six sub-themes (occupational risk associated with sex work, increased chance of offering unprotected sex, positive encouragement from others, need to take care of children, prior participation in HIV prevention research studies and GBV). While intrinsic motivation refers to FSWs taking PrEP because they find it rewarding in itself, extrinsic motivation is driven from an external force with an expectation to receive some perceived rewards [30]. All names used in this section, are pseudo names chosen by FSW to ensure anonymity.

**Intrinsic motivational factors for FSWs initiating PrEP.** Intrinsic factors for initiating PrEP among FSWs refer to data depicting participants' personal desires to remain HIV

**Table 1. Demographic table.**

| Item | Sub-category | N = 20 | Percentage (%) |
|---|---|---|---|
| Age in years | | | |
| | 18–24 | 4 | 20% |
| | 25–35 | 9 | 45% |
| | 36–45 | 6 | 30% |
| | 46–55 | 1 | 5% |
| Educational level | Primary school | 4 | 20% |
| | Secondary school | 11 | 55% |
| | College level | 5 | 25% |
| Marital status | Separated/divorced | 15 | 60% |
| | Never married | 4 | 20% |
| | Cohabiting | 1 | 5% |
| Number of children | No children | 2 | 10% |
| | One child | 3 | 15% |
| | Two children | 6 | 30% |
| | Three children | 6 | 30% |
| | Four children | 2 | 10% |
| | More than four children | 1 | 5% |

negative by initiating on PrEP. These included desire for self-protection from HIV infection and fear for one's health, fear of getting infected with HIV and condoms bursting.

**Self-protection from HIV infection.** From the data, it emerged that participants reported a strong desire to preserve their health and protect themselves from HIV infection with most living in constant fear for their health. This was a strong motivation to initiate PrEP. This is illustrated in the following quotes from FSWs.

*I saw PrEP as something good considering the work that l am into. When we meet our clients, sometimes the condoms burst, so l realised the importance of taking PrEP so that l am protected. . .. In our line of work, we meet various clients so personally l would want to use condoms but at the same time l should be taking PrEP to protect myself from HIV. (Lollipop, P4, 25–35 years)*

*What made me start PrEP is that I am a sex worker so most of the time I meet clients who do not like the condom. So starting PrEP has now made me fearless when I have sex. Let us say, when I have sex with a person, right, we wear a condom for protection then it bursts. I am not that afraid because I will know that on my side, I am protected. (Tess, P15, 18–24 years)*

*What made me to go on PrEP is that, because of my age, I am now at 56 years old, l thought it is better to have some kind of protection, being someone who has survived up to this day. He may come infected or he may come not infected. I have reached this far without contracting HIV, I can't afford to get HIV now. . . (Nyarai, P10, 56+ years)*

*I realized that if I take PrEP maybe I can safeguard my life since I heard that PrEP prevents you from contracting the AIDS virus. (Laura, P11, 25–35 years)*

**Fear of getting infected with HIV.** One recurrent idea from the data was that of participants living in constant fear of contracting HIV as a motivator to initiate PrEP as demonstrated below:

*". . .As sex workers, we meet different clients and we do not know their HIV status so, it is better for us to be on PrEP and protect ourselves". (Tess,P15, 18-24years)*

*(Pause), because of this job that l am into (sex work), l had to get tested. l first got tested here at the Clinic. When my first results came out stating that I am HIV negative, l actually did not believe it. Then l went to another clinic and I was tested again. The same results came out. When the mobile testers with tents came, l went in again and the same result came out. That's when l came back here and l was initiated on PrEP because I knew that I could be infected any time. (Nokutenda P18, 25–35 years).*

**Condoms bursting.** One frequently occurring concern from the FSWs was the condoms bursting due to their clients being drunk or using force during sex. This issue is exemplified by the following quotes;

*We meet a lot of different clients and you find that at times condoms burst.., (Tsitsi, P1, 36-45years)*

*Some of our clients can be very violent such that the condom itself may end up breaking. In such cases, if you are on PrEP at least you know you are protected (Nyasha P5, 36-45years)*

**Extrinsic motivational factors for FSWs initiating PrEP.** Extrinsic motivational factors of FSWs initiating on PrEP refers to the external forces that participants reported driving them to initiate on PrEP. These factors were occupational risk associated with sex work, increased chance of offering unprotected sex, positive encouragement from others, gender-based violence (GBV) the need to take care of children and prior participation in PrEP research studies.

**Occupational risk associated with sex work.** The first emerging sub-theme in this category of external motivators to initiating PrEP was the risk associated with sex work as a trade. This is exemplified in the voices of the participants below:

> It took me about a week to decide whether l should take this (PrEP) or not, but l guess my job as a sex worker pushed me to initiate on PrEP. (Natasha, P19, 25–35 years)

> We are sex workers, so our risk is high. So, it (PrEP) works as a barrier to getting HIV in that, our risk to get this disease is high. If I can encounter a burst condom or anything like that, at least by having something like PrEP, it is protecting me. (Nyasha, P5, 25–35 years)

> I realized the importance of PrEP in my job (sex work). I was interested because when I am faced with emergencies that occur, like at times a condom might burst. I just said, let me just take PrEP so that I can be protected. (Mercy, P12, 25–35 years)

**Increased chance of offering unprotected sex as a motivator to initiate PrEP.** Increased financial freedom as a motivator to initiate PrEP emerged as a strong theme in this category of extrinsic motivational factors of sex workers initiating PrEP. Some of the younger FSWs viewed sex work as a temporary measure to meet their financial needs as echoed by the extracts from the participants' voices below:

> What made me take PrEP is my work. I also wanted to raise enough money to start my business quickly, because when you have unprotected sex, you can charge more money.

> When I leave sex work, I want to still be healthy and PrEP will help me achieve that, so being on PrEP will increase my chances for competing and getting more clients who want unprotected sex. (Melody, P13, 25–35 years)

> Sometimes l know that my client prefers unprotected sex and pays well. So, as a sex worker, l have a sweet tongue. I will sweet talk him that l will put on the female condom since he does not want to put on one. That way, l will put on the female condom. If he still refuses, I raise the price, knowing that at least I am safe because of PrEP. (Nokutenda P18, 25–35 years)

> Our clients can be funny, they pay more if you have unprotected sex with them. They say, who wants to eat a sweet in a paper. PrEP will help reduce anxiety around HIV infection in cases of unprotected sex. (Chrystal, P17, 25–35 years)

**Positive encouragement from others.** This theme was based on participants reporting receiving some positive encouragement from other people such as fellow sex workers, friends and family members as shown below:

> I discussed PrEP with my mother, who is on ART and she (my mother) taught me that there is a difference between PrEP and ART. She is the one who actually encouraged me to take PrEP. She told me that it will protect me from getting sick like her. She contracted HIV from this

*work (sex work), so we don't have to both die from the same thing (HIV), because we need the money. (Natasha P19, 25–35 years)*

*She is the one who told me that at the clinic you can get PrEP. When the outreach workers came to our community, that is when I started going to get PrEP. (Laura, P11, 25–35 years)*

*Ah, let me just say, where we go to look for money, the girls there always talk about it (PrEP) and how it helps them stay safe and get more clients. So, when I came here on Tuesday after I had been seen by the nurse, I just decided to take it (PrEP). (P13, Melody, 25–35 years)*

**Prior participation in PrEP research studies.** Interestingly, data showed that the participants who had previously participated in PrEP clinical trials were somehow motivated through the studies to initiate PrEP in our research setting when the clinical trials ended, as echoed in the quotes below:

*l first enrolled in a study called ASPIRE (an HIV prevention clinical trial on the vaginal Ring), later on l went onto HOPE. When both studies came to an end, my boyfriend, who is HIV positive, was now staying with me (at my house) permanently. So, when the study ended, l just saw it fit that l should go on PrEP because of his cruelty to me. Sometimes he would boil my family planning pills and hide my female condoms. (Fortu, P2, 36–45 years)*

*I participated in ASPIRE (an HIV prevention clinical trial) and when that study ended, I wanted to continue protecting myself. I then initiated on PrEP because I believe that PrEP protects one from HIV. We learnt that from the study. (Laura, P11, 25–35 years)*

**Gender Based Violence (GBV).** Gender Based Violence is defined as any violence against a certain gender that result in or is likely to result in physical, sexual or psychological harm or suffering to that gender [31] Some participants reported experiences of GBV as some reside with their partners who were living with HIV, hence the need to protect themselves through PrEP. Gender Based Violence was reportedly experienced by some FSWs as a result of being on PrEP. One FSW reported that;

*My boyfriend would beat me up saying he did not want to see me taking these pills (PrEP) as they would encourage me to sleep with other men (Tsitsi P1, 36–45 years).*

*Some of our clients are very cruel. At times we are raped and we have no opportunity to use condoms, so, being on prep will protect us from HIV (Mercy, P12, 25–35 years).*

**Need to take care of the children.** Most FSWs who participated in the study who and who had children said they were motivated to initiate PrEP so that they can take care of their children, as expressed in the following participants' voices below:

*When it comes to PrEP, I liked it and I was initiated on PrEP. Of cause, they say it is only 98% effective, but I cannot get infected with the disease (HIV). I can now take care of my children, that's my wish, till my kids are done with school then see my grandchildren because if I fall sick, I do not have a younger sister, I do not have a brother to look after my family when I am gone. I had to protect myself. (Chipo, P3, 26–35 years)*

*I do this work because l have a family. l should be able to look after my children and my work should just continue while l am healthy. At least PrEP will help me achieve that. I won't get the disease and leave them motherless. (Mercy, P12, 25-35years)*

*You see, the kind of risks that we come across as we meet with our clients are many. Like you can have a condom bursting. So, we should always use PrEP as a way of protecting ourselves for the sake of our children. There is no sex worker who does not have a child. Most of us have children that still need to be cared for. So, we cannot leave ourselves to just accept an infection yet we have the chance to get PrEP. (Lollipop P4, 25–35 years).*

*Personally, when l decided to take PrEP, l saw it as something good considering the work that l am doing. . ..., l realised the importance of taking PrEP so that l am protected from HIV. It is also because l have a family, l should be able to look after my children, and my work should just continue while l am healthy. In our line of work, we meet various clients, so personally l would want to use condoms but at the same time, l should be taking PrEP. So, that way it will be safe for me, l do not see anything wrong in starting to take PrEP. (Mukaranga, P20, 25–35 years).*

These findings show both internal and external drivers that motivate FSWs to initiate PrEP in Harare, Zimbabwe.

## Discussion

The objective of the study was to identify factors that motivate FSWs to initiate PrEP using DPA as the methodology. The findings revealed two main themes namely intrinsic and extrinsic motivational factors. Intrinsic factors included self-protection from HIV infection and condoms bursting whilst extrinsic motivators included occupational risk associated with sex work, increased chances of offering unprotected sex, positive encouragement from others, prior participation in PrEP studies, the need to take care of children and GBV. This descriptive phenomenological study helped in identifying intrinsic and extrinsic drivers that motivate Female Sex Workers (FSWs) to initiate Pre-Exposure Prophylaxis (PrEP) in Harare, Zimbabwe. Our study fills an evidence gap in literature addressing FSW motivation for PrEP initiation and is a response to the World Health Organization (WHO) (2015) recommendations of having PrEP offered to population groups with an HIV incidence of 3% or higher [4]. FSWs in Zimbabwe have an estimated HIV incidence of 10% [31]. Whilst there is literature on motivations for PrEP initiation among other population groups, such as, SDC, PBFW, MSM, GBM and the general population, there is paucity of evidence for FSWs [9–13,15]. This makes it difficult to develop appropriate strategies for promoting PrEP use and PrEP adherence among FSWs [32].

Broad FSW motivators included, self-protection from HIV and access to research through participation in HIV prevention clinical trials. We also found specific motivators, peculiar to FSWs which included, financial freedom to charge higher rates for unprotected sex and the need to take care of their children. In Zimbabwe, more than 1.3 million children are orphaned to HIV [2], including many of the sex workers themselves. These FSWs do not wish the same fate for their children.

Our findings suggest that factors motivating FSWs to initiate PrEP include both intrinsic and extrinsic motivational factors. Despite the WHO guidance recommending all populations with an HIV incidence of 3% or higher be prioritized for PrEP, PrEP initiation requires further motivation beyond the elevated HIV risk perception [2,33,34].

Violence, including gender-based violence, emerged as a motivator for initiating PrEP. Violence has been associated with HIV risk across different population groups. This finding is

consistent with a study among pregnant and breast-feeding women which reported forced sex and GBV as motivational factors for prep initiation [35]. In a study among men who have sex with men (MSM) who reported experiencing physical, emotional and sexual violence frequently, violence was a major motivator for MSM to initiate PrEP to ensure they are protected from HIV [8].

The study findings also suggest that FSWs experience motivators that may be unique to them as part of their work. Some of the PrEP motivational factors which may be unique to FSWs included positive encouragement from other FSWs and having increased chances of offering unprotected sex as they will feel they are protected from HIV through PrEP. These motivational factors suggest the need to develop psychosocial interventions to better support FSWs to initiate PrEP.

Our findings highlight that the FSWs are aware that they need to stay healthy to enable them to sustain their livelihood. They are acutely aware of the risks associated with their trade, where men promise to pay more to have unprotected sex as well as the imminent dangers of condoms bursting during sex. This has heightened FSW need for PrEP.

Post-trial access is a huge ethical concern. When FSWs volunteer to participate in HIV prevention clinical trials, researchers have an ethical obligation to ensure the women have access to the efficacious HIV prevention products beyond the trial to ensure they continue to protect themselves against HIV [2,36]. Our findings reveal that some FSWs transition from one HIV prevention trial to the next to enable access to HIV prevention. Clinical trials were reported to deliver free, quality services, with clients reporting to be treated respectfully compared to public health facilities. UNAIDS calls for the protection of all persons participating in clinical HIV prevention clinical trials and recommends that all clinical trials ensure there is post-trial access for anyone participating in clinical trials [2]. This was evidently not the case with the FSWs participating in one of the previous PrEP trials conducted in Zimbabwe, who reported resorting to accessing PrEP at the specialized clinic after they had participated in the HIV prevention trials.

Whilst there may be some similarities in motivational factors for PrEP initiation among FSWs and other population groups [9–13,15], including sero-discordant couples (SDCs) whose motivation for PrEP is to remain HIV negative whilst preserving their relationships [35,37], it must be noted that FSWs are a unique population group that require tailor-made strategies and interventions to ensure approaches are amplified and are delivered on a larger scale for population-level impact [36]. It is critical that our findings be aligned to PrEP demand creation and awareness campaigns by drawing from both internal and external motivators. We recommend that these findings be considered in the development of all forms of PrEP awareness campaigns and PrEP literacy materials in the media, including both print and electronic channels targeting FSWs [38,39]. Given the similarities in HIV risk between FSWs and SDCs, some of the PrEP demand creation materials and messaging may equally apply to both groups as the overriding factor is to remain HIV negative.

## Limitations

This study was conducted soon after the participants had been initiated on PrEP and therefore there is no way of knowing whether the FSWs would stay motivated if they were to experience side effects.

## Conclusions

Understanding the factors that motivate FSWs to initiate PrEP is critical in developing contextually appropriate strategies to promote PrEP initiation and adherence strategies within

specific and eligible populations for receiving PrEP according to the WHO guidelines (2015) [40]. The study highlighted broad FSW intrinsic motivators such as self-protection from HIV and fear for one's health. Extrinsic motivators included occupational risk associated with sex work, financial freedom through higher prices charged for unprotected sex, positive encouragement from others, the need to take care of children and prior participation in PrEP research studies. Policy makers and programmers need to take these critical factors into account to better engage FSWs in PrEP and ultimately contribute towards ending AIDS by 2030 [41].

## Supporting information

**S1 File.**
(XLSX)

## Author Contributions

**Conceptualization:** Definate Nhamo, Frances M. Cowan.

**Data curation:** Definate Nhamo.

**Formal analysis:** Definate Nhamo, Sinegugu E. Duma.

**Investigation:** Definate Nhamo.

**Methodology:** Definate Nhamo, Sinegugu E. Duma, Dixon Chibanda, Frances M. Cowan.

**Project administration:** Definate Nhamo.

**Supervision:** Sinegugu E. Duma, Dixon Chibanda, Frances M. Cowan.

**Validation:** Definate Nhamo.

**Visualization:** Definate Nhamo.

**Writing – original draft:** Definate Nhamo.

**Writing – review & editing:** Elizabeth B. Ojewole, Dixon Chibanda, Frances M. Cowan.

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
