## [Decision Letter · Decision Letter 0]

28 Mar 2022

PONE-D-22-04107Motivation for Initiating Pre-exposure Prophylaxis (PrEP) for HIV Prevention among Zimbabwean Female Sex workers.PLOS ONE

Dear Dr. Definate Nhamo,

Thank you for submitting your manuscript to PLOS ONE. After careful consideration, we feel that it has merit but does not fully meet PLOS ONE’s publication criteria as it currently stands. Therefore, we invite you to submit a revised version of the manuscript that addresses the points raised during the review process.

We look forward to receiving your revised manuscript.

Kind regards,

Hamid Sharifi

Academic Editor

PLOS ONE

Journal Requirements:

Reviewers' comments:

Reviewer's Responses to Questions

**Comments to the Author**

1. Is the manuscript technically sound, and do the data support the conclusions?

Reviewer #1: Yes

Reviewer #2: No

Reviewer #3: Yes

2. Has the statistical analysis been performed appropriately and rigorously? 

Reviewer #1: Yes

Reviewer #2: No

Reviewer #3: Yes

3. Have the authors made all data underlying the findings in their manuscript fully available?

Reviewer #1: Yes

Reviewer #2: No

Reviewer #3: Yes

4. Is the manuscript presented in an intelligible fashion and written in standard English?

Reviewer #1: Yes

Reviewer #2: Yes

Reviewer #3: Yes

5. Review Comments to the Author

Reviewer #1: This is a simple, straightforward , qualitative ,well written research paper on intrinsic and extrinsic factors that motivate women in sex work to start PrEP. It uses the seven steps of Colaizzi's method for data analysis which is a well known method used in qualitative research. I really don't have anything new to add to this paper other than saying that

since this was a small group of women (20) the findings and experiences could have been corroborated further with a Focus Group Discussion, but I guess this was not possible as the topic of sex work is a sensitive one and

requires confidentiality.

Some corrections need to be done in this paper as follows :

1. Line 33 - the first point for intrinsic factor is expressed clearly through a (i). The second point -fear for one's health

should also be preceded by (ii)

2.Line 106- The word illicit should be replaced by elicit.

3. Line 231- the first word in this line is not clear.

4.Lines 316-320 and lines 330 -333 are identical to each other though used in different context.

5.Table -1 The item column should be clearly elucidated. n should be numbers, and percentage followed by % symbol.

The supplementary data is available.

Reviewer #2: Dear Editor,

Thank you for providing me the opportunity to review this manuscript. There are many concerns which are as fallow:

Title:

1) The title is not reflecting the aim of the study.

2) It is strongly suggested to consider the study design in the title.

Abstract:

1) The aim is “to understand factors that motivate FSWs to initiate PrEP, in Harare, Zimbabwe” while the tile seem is about if participants are motivated or not.

2) This sentence needs English edit “Data was analyzed using Colaizi’s seven steps to analyze DPA data.”.

3) In the results section, it has mentioned that 7 themes were extracted, according to my knowledge the themes should be broader and more limited in number. It seems these are subthemes.

Key word:

1) there was no keyword.

Introduction:

1) Lines 47- 48, the reference 3 is not cited in the text.

2) There is no explanation about the context.

Method:

1) While estimating sample size for qualitative research is mentioned in some literature, however it is not enough and it will be finalized by data saturation. Please explain about it.

2) Line 120, why it has referenced to other study?

3) Only one month has dedicated to sampling? How it is possible? In the qualitative research the sampling and data analysis are conducted simultaneously to better manage the interview process and the study. It seems that the authors interviewed all participants and then analyzed the data.

4) Who were responsible for interview and how she/he were trained for interviewing?

5) Line 139, why it has referenced to other studies?

6) Please report the interview guide.

7) Please move “Member checking involves the taking of findings back to study participants to ensure data has been interpreted correctly taking account of the context” to the rigor section.

8) Please explain about external validity and maximum variation.

9) Please explain about the time of data analyzing, and the software which has used for data managing.

10) Please report an example for the data analysis process.

Results:

1) I believe the authors has wrongly used the word “theme”.

2) “Fear of getting infected with HIV and condoms bursting” is not reported in the abstract.

3) I see no different between the “Self-protection from HIV infection and fear for one’s health” and “Fear of getting infected with HIV” subthemes.

4) according to the quotations, I believe “Increased financial freedom as a motivator to initiate PrEP” could be replace with “Increased the chance for offering unprotected sex”.

5) The “Gender based Violence (GBV)” is not well explained and even though the quotation is no related.

“When both studies came to an end, my boyfriend, who is HIV positive, was now staying with me (at my house) permanently. So, when the study ended, l just saw it fit that l should go on PrEP because of his cruelty to me. Sometimes he would boil my family planning pills and hide my female condoms. (Fortu, P2, 36-45 years)”!!

6) I believe the results are not rich.

Discussion

1) Line 367- 368, why it has referenced to other studies?

2) The first paragraph of the discussion is a repetition of introduction.

3) The discussion should be more concise and need more interpretation and comparing with evidence.

4) When we seek the participants experience it is not related when he/she has experienced it. On the other had it is a major concern of the present study which did not considered different experience and did not consider maximum variation in sampling.

Reviewer #3: No additional comments. The authors presented findings from a well designed and conducted study with 20 FSWs. The findings will be informative for individuals and programs responsible for developing and implementing messages and counseling efforts that meet the PrEP access needs of FSW. From a public health perspective I which surveys were also conducted with FSWs who have either not sought out PrEP or refused to take it, as those findings would be equally if not more important in program design.

6. PLOS authors have the option to publish the peer review history of their article (what does this mean?). If published, this will include your full peer review and any attached files.

Reviewer #1: No

Reviewer #2: **Yes: **Mahlagha Dehghan

Reviewer #3: No

---

## [Author Response · Author response to Decision Letter 0]

18 Apr 2022

Thank you for your comments. We have addressed each and everyone of the comments raised by the two reviewers in our "Response to Reviewers" document. We have alas tracked all the changes we made in response to the comments in the tracked changes version of the manuscript. We have also provided a clean version of the manuscript after accepting all tracked changes. 

Thank you once again for all the comments.

---

## [Editor Report · Decision Letter 1]

21 Apr 2022

PONE-D-22-04107R1Factors motivating female sex workers to initiate Pre-exposure Prophylaxis for HIV Prevention in ZimbabwePLOS ONE

Dear Nhamo

Thank you for submitting your manuscript to PLOS ONE. After careful consideration, we feel that it has merit but does not fully meet PLOS ONE’s publication criteria as it currently stands. Therefore, we invite you to submit a revised version of the manuscript that addresses the points raised during the review process.

Dear Authors,

The uploaded files are disorganized and this makes difficult for the editor and reviewers to check them. Please confirm the files are uploaded in a organized way before final submission. Also, use the comments and responses in two colors to make it easier for the reviwers to check them.

We look forward to receiving your revised manuscript.

Kind regards,

Hamid Sharifi

Academic Editor

PLOS ONE

Journal Requirements:

Additional Editor Comments (if provided):

Dear Authors,

The uploaded files are disorganized and this makes difficult for the editor and reviewers to check them. Please confirm the files are uploaded in a organized way before final submission. Also, use the comments and responses in two colors to make it easier for the reviwers to check them.

---

## [Author Response · Author response to Decision Letter 1]

5 May 2022

Dear editor,

I am confirming that the files have been uploaded in an organized way. We have the following attachments:

1. Letter to the editor

2. Response to Reviewers'

3. Manuscript

4. Revised Manuscript with Track Changes

Regards,

Definate Nhamo (on behalf of all co-authors)

---

## [Decision Letter · Decision Letter 2]

26 May 2022

PONE-D-22-04107R2Factors motivating female sex workers to initiate Pre-exposure Prophylaxis for HIV Prevention in ZimbabwePLOS ONE

Dear Dr. Definate Nhamo,

Thank you for submitting your manuscript to PLOS ONE. After careful consideration, we feel that it has merit but does not fully meet PLOS ONE’s publication criteria as it currently stands. Therefore, we invite you to submit a revised version of the manuscript that addresses the points raised during the review process.

Dear AuthorsPlease revise the file based on the reviewer's comments.

We look forward to receiving your revised manuscript.

Kind regards,

Hamid Sharifi

Academic Editor

PLOS ONE

Reviewers' comments:

Reviewer's Responses to Questions

**Comments to the Author**

1. If the authors have adequately addressed your comments raised in a previous round of review and you feel that this manuscript is now acceptable for publication, you may indicate that here to bypass the “Comments to the Author” section, enter your conflict of interest statement in the “Confidential to Editor” section, and submit your "Accept" recommendation.

Reviewer #1: All comments have been addressed

2. Is the manuscript technically sound, and do the data support the conclusions?

Reviewer #1: Yes

3. Has the statistical analysis been performed appropriately and rigorously? 

Reviewer #1: N/A

4. Have the authors made all data underlying the findings in their manuscript fully available?

Reviewer #1: Yes

5. Is the manuscript presented in an intelligible fashion and written in standard English?

Reviewer #1: Yes

6. Review Comments to the Author

Reviewer #1: the paper is well written and explained in great detail.

some corrections :

1. Line 32 under abstract - the line should read intrinsic and extrinsic factors .

2. Line 96- when expanding DPA for the first time , the first letter of each word should be in capitals

3. Table one - please add age in years under item

4. Line 406- Several abbreviations are mentioned -kindly expand

7. PLOS authors have the option to publish the peer review history of their article (what does this mean?). If published, this will include your full peer review and any attached files.

Reviewer #1: No

---

## [Author Response · Author response to Decision Letter 2]

27 May 2022

Dear Editor, 

Please find in the table below our individual responses to the comments raised by reviewer 1. 

Reviewer #1: the paper is well written and explained in great detail.

some corrections :

1. Line 32 under abstract - the line should read intrinsic and extrinsic factors .

2. Line 96- when expanding DPA for the first time , the first letter of each word should be in capitals

3. Table one - please add age in years under item

4. Line 406- Several abbreviations are mentioned -kindly expand

Specific responses to comments raised by reviewer 1:

Comment 1: Line 32 under abstract - the line should read intrinsic and extrinsic factors. Thank you for the comment. We have added ”intrinsic and extrinsic factors”.

Comment 2:Line 96- when expanding DPA for the first time , the first letter of each word should be in capitals. Thank you for the comment, we have changed this to “Descriptive Phenomenological Analysis”.

Comment 3:Table one - please add age in years under item. Thank you, we have added “Age in years” under item.

Comment 4:Line 406- Several abbreviations are mentioned -kindly expand. Thank you, we have expanded the abbreviations as follows;

Pre-Exposure Prophylaxis (PrEP)

Female Sex Workers (FSWs)

The World Health Organization (WHO) 

Yours Sincerely

Definate Nhamo

---

## [Editor Report · Decision Letter 3]

30 May 2022

Factors motivating female sex workers to initiate Pre-exposure Prophylaxis for HIV Prevention in Zimbabwe

PONE-D-22-04107R3

Dear Dr. Definate Nhamo

We’re pleased to inform you that your manuscript has been judged scientifically suitable for publication and will be formally accepted for publication once it meets all outstanding technical requirements.

Kind regards,

Hamid Sharifi

Academic Editor

PLOS ONE
---

## [Editor Report · Acceptance letter]

14 Jun 2022

PONE-D-22-04107R3 

Factors motivating female sex workers to initiate Pre- Exposure Prophylaxis for HIV Prevention in Zimbabwe 

Dear Dr. Nhamo:

I'm pleased to inform you that your manuscript has been deemed suitable for publication in PLOS ONE. Congratulations! Your manuscript is now with our production department. 

Kind regards, 

on behalf of

Dr. Hamid Sharifi 

Academic Editor

PLOS ONE